# Spinal Cord Injury and Loss of Cortical Inhibition

**DOI:** 10.3390/ijms23105622

**Published:** 2022-05-17

**Authors:** Bruno Benedetti, Annika Weidenhammer, Maximilian Reisinger, Sebastien Couillard-Despres

**Affiliations:** 1Institute of Experimental Neuroregeneration, Paracelsus Medical University, 5020 Salzburg, Austria; bruno.benedetti@pmu.ac.at (B.B.); annika.weidenhammer@meduniwien.ac.at (A.W.); m.reisinger@stud.pmu.ac.at (M.R.); 2Spinal Cord Injury and Tissue Regeneration Center Salzburg (SCI-TReCS), 5020 Salzburg, Austria; 3Austrian Cluster for Tissue Regeneration, 1200 Vienna, Austria

**Keywords:** cortical inhibition, spinal cord injury, neocortex, disinhibition, interneuron, transcranial magnetic stimulation, TMS

## Abstract

After spinal cord injury (SCI), the destruction of spinal parenchyma causes permanent deficits in motor functions, which correlates with the severity and location of the lesion. Despite being disconnected from their targets, most cortical motor neurons survive the acute phase of SCI, and these neurons can therefore be a resource for functional recovery, provided that they are properly reconnected and retuned to a physiological state. However, inappropriate re-integration of cortical neurons or aberrant activity of corticospinal networks may worsen the long-term outcomes of SCI. In this review, we revisit recent studies addressing the relation between cortical disinhibition and functional recovery after SCI. Evidence suggests that cortical disinhibition can be either beneficial or detrimental in a context-dependent manner. A careful examination of clinical data helps to resolve apparent paradoxes and explain the heterogeneity of treatment outcomes. Additionally, evidence gained from SCI animal models indicates probable mechanisms mediating cortical disinhibition. Understanding the mechanisms and dynamics of cortical disinhibition is a prerequisite to improve current interventions through targeted pharmacological and/or rehabilitative interventions following SCI.

## 1. Introduction: SCI Harms the Brain

Traumatic SCI is a sudden and unpredictable incident that destroys portions of the spinal cord, leading to motor and sensory deficits, as well as dysfunctions of the somatic and autonomic nervous systems [1]. Beyond the loss of movement control, typical deficits include the loss of bladder and bowel control, declined sexual functions and chronic pain, among others [1,2]. SCI can occur at any age, and the damage is irreversible. However, constant improvements in healthcare and treatment, as well as increased awareness about the needs of patients over the last century, have significantly ameliorated the quality of life and lifespan following SCI [3,4,5,6,7,8,9]. Thus, it is even more pressing to identify interventions enabling the recovery of functions lost after SCI. The recovery of muscle control is a crucial element to improve the quality of life and the autonomy of SCI patients. Accordingly, rehabilitation and active lifestyle have been recognized as crucial processes that help to regain independence and to reduce health complications resulting from prolonged inactivity [10,11,12,13,14]. Nevertheless, the timely implementation of efficient strategies remains often neglected, affecting motor recovery and, together with accompanying morbidities, decreasing the likelihood of returning to a fully independent life routine [15,16]. Furthermore, various therapies addressing the symptoms of SCI are being developed with promising outcomes for management and reduction in secondary damage, increased neuroprotection and improved neuroregeneration [5,17]. However, despite constant improvements, an effective cure, leading to major functional recovery based on the regeneration of neuronal connectivity across the lesion, is still missing [17,18,19,20].

Many of the neurons that become disconnected following SCI reside outside the spinal cord, such as the motor neurons of the primary motor cortex, which are crucial for the control of voluntary movements. These disconnected neurons are a resource for the long-term regeneration and functional recovery of the central nervous system. However, the axotomy resulting from SCI has an impact on the physiology of the cortical and corticospinal network [21,22,23], which can complicate or even hinder the recovery process. Several attempts to address the clinical symptoms of SCI have therefore explored the possibility to retune neuronal activity in the corticospinal network [24,25]. Finding ways to reconnect cortical motor neurons to their original targets constitutes a daunting task. In addition, early assessments of the severity of SCI, especially in an acute situation, are difficult and inherently inaccurate [26]. This lack of knowledge is a major hurdle for the design of an effective and patient-specific treatment. Therefore, the assessment of cortical activity in SCI patients, for example, using electro-encephalography (EEG), transcranial magnetic stimulation (TMS), etc., has been extremely convenient, as it relies on non-invasive techniques [27,28]. In addition, animal models are available to resolve the molecular mechanisms of brain dysfunction after SCI. In this review, we take advantage of the complementarity of clinical research and basic research to offer a multifaceted overview about cortical network dysfunction after SCI and putative underlying mechanisms.

## 2. TMS as a Method to Analyze the Loss of Inhibition after SCI

Following SCI, extensive functional plasticity and the reorganization of neuronal circuits often involve altered inhibitory neurotransmission [29,30,31,32,33,34,35,36,37]. Although brain disinhibition has been explored extensively for pathologies such as peripheral deafferentation and stroke [38,39,40,41,42,43], it remains resolved to a lesser extent in the context of SCI [44]. The establishment of direct comparisons between cortical output and motor performance after SCI and after other pathologies, such as peripheral deafferentation or stroke, is hindered by discrepancies intrinsic to various conditions. Therefore, this review focuses on the phenomenon of cortical disinhibition as a central component of the pathophysiology of SCI. Even under such focused premises, disinhibition appears as a startlingly heterogeneous process that needs to be first disentangled to be considered in the design of therapy and treatment following SCI.

Here, we consider the specific patterns of functional cortical derangement after SCI, focusing primarily on works based on TMS. TMS is a technique that evokes neuronal activity by the juxtaposition of magnetic coils to the skull and the generation of magnetic fields, delivered as pulses, in selected and focalized cortical areas. Therewith, evoked electrophysiological activity can be measured from the cortex, spinal cord and muscles targeted by the stimulated motor areas, and specific parameters in the readout reflect the relative excitability and inhibition of cortical and corticospinal circuits [45,46]. However, TMS has limitations, including the poor resolution of direct and indirect stimulus effects, ambiguities in the causal relationship of therapeutic TMS and behavioral improvement, and caveats when combining TMS with some other analytical techniques [47,48]. Most relevantly, the reliability of TMS metrics varies depending on the residual muscle strength of the tested muscle, which, if not accounted for, can significantly affect measurements assessing recovery from SCI [49]. Nevertheless, TMS is useful to stimulate the brain safely and non-invasively in awake patients and to measure specific brain activities and functional alterations after SCI [50,51,52,53]. Therefore, decades of work based on TMS have successfully outlined the traits of altered brain physiology after SCI (Figure 1) [51,54,55,56].

## 3. Loss of Inhibition Promotes Motor Recovery after SCI

Early works have suggested that the alteration of inhibitory mechanisms ensues from SCI and causes increased excitability and plasticity in the central nervous system [34,58]. Accordingly, reduced thresholds for effective TMS of muscles innervated by segments above the lesion site were measured in patients with complete thoracic SCI, and this observation was interpreted as evidence for hyper-excitability and remodeling after injury [59]. The loss of inhibition and network remodeling was also suggested by experiments involving paired-pulse TMS [60]. Namely, increased Indirect-wave (I-wave) facilitation was detected after SCI, whereas the motor conduction time remained unaltered. In this context, I-waves originate from the TMS-evoked output of cortical excitatory neurons, which are regulated by a network of GABAergic interneurons [66,67]. Thus, it was proposed that increased I-wave facilitation after SCI results from a loss of cortical inhibition. The modulation of I-waves resulting from dynamic interaction of the excitatory and inhibitory network is relevant to determine the extent of spinal motor neuron activation [68,69,70,71]. In this scenario, cortical hyper-excitability resulting from disinhibition may therefore strengthen the output of spared fibers during motor recovery. Thus, the loss of inhibition can, in this view, ameliorate the cortical output, strengthen motor control and improve motor recovery [72].

Independent evidence converging towards a similar conclusion, i.e., the occurrence of the loss of inhibition after SCI, was obtained by measuring cortical silent periods (CSP). A CSP indicates the duration of the transient decrease in motor neuron excitability after a TMS pulse. SCI was found to shorten the CSP [57], which can be regarded as a consequence of disinhibition and a contribution of cortical remodeling to facilitate motor performance recovery [57,73]. In line with these studies, a clinical study supporting cortical remodeling and disinhibition after SCI has shown that the peripheral stimulation of lower limbs can prime contextual hand flection in SCI patients [44]. A more recent work agrees with the occurrence of disinhibition after SCI [61]. In this case, the experimental readout was based on measurements of the resting motor threshold and motor evoked potentials (MEP) upon TMS. Thereby, Nardone and colleagues observed that the input/output ratio (i.e., stimulation intensity/MEP) decreased in individuals with SCI, whereas the resting motor threshold remained constant. Based on previous works, the authors have discussed that the combined readout associates with increased excitability of spinal rather than cortical areas [66,74,75,76,77,78,79]. The observed alteration in motor output is therefore caused by the remodeling and loss of inhibition, causing the activation of a larger fraction of the pool of excitatory synapses for a given stimulus intensity. Once more, disinhibition is portrayed as a rescuing effect contributing to motor recovery.

## 4. The Loss of Inhibition Aggravates Symptoms after SCI

The works cited so far suggest that disinhibition and increased CNS excitability are beneficial for motor recovery after SCI. However, there is evidence to suggest that cortical disinhibition is detrimental: the exacerbated loss of cortical inhibition and hyper-excitability can lead to maladaptive plasticity and neuropathic pain [80,81]. For example, suppressed inhibition and damage to the thalamocortical network can lead to neuropathic pain after SCI [2]. Additional manifestations associated with the loss of central nervous system inhibition after SCI are “referred phantom sensations” [62]. In this context, repetitive TMS was used to treat referred phantom sensations, and it was proposed to restore cortical inhibition through the reactivation of silent intracortical interneurons. Moreover, the authors studying reduced short-interval intracortical inhibition after SCI argued in favor of an inhibitory impairment. The latter contributes to defective corticospinal control, hindering the accurate planning and/or termination of voluntary muscle contractions, therefore causing poor motor control [82,83].

In summary, a complex clinical scenario reveals that the loss of inhibition following SCI can be helpful on one side to intensify the output signal of cortical motor neurons to the spinal cord. On the other hand, excessive disinhibition can also contribute to the development of chronic pain and maladaptive plasticity, and it can hinder motor control [81]. Since disinhibition seems to have both pros and cons, it should be considered whether a treatment that alters the degree of cortical inhibition would rather cause harm or benefit. A better resolution over the mechanisms and temporal patterns of disinhibition can be crucial for personalized choices on therapeutic options.

## 5. Alternative Mechanisms and Explanations for Imbalanced Cortical Excitability

Even though the loss of CNS inhibition is often observed after SCI, it may not be a condition that applies to all patients [64] nor fully account for CNS plasticity. Indeed, some reports describing increased CNS excitability after SCI did not conclude a loss of inhibition and therefore did not consider retuning the inhibitory network as a putative therapeutic strategy [52,59,84]. For instance, paired stimulation of peripheral nerve and cortical areas promotes corticospinal transmission through a form of plasticity which does not involve the modulation of inhibitory networks [85]. Moreover, paired associative stimulation has been shown to improve motor recovery [86] and is associated with long-term potentiation of excitatory neurotransmission [86,87] that has been suggested to involve the structural reorganization of the corticomotoneuronal synapses of the cervical spinal cord. Finally, TMS after SCI has been reported to directly increase glutamate receptor activation and strengthen excitatory neurotransmission [88]. Therefore, increased excitatory neurotransmission may also be possible independently of processes of disinhibition.

As a consequence, although not always present, disinhibition may well occur concomitantly with other plasticity mechanisms, resulting in apparent heterogeneity and discrepancy between experimental outcomes [50,63,89]. Furthermore, some works have provided evidence against cortical disinhibition and in favor of increased corticospinal inhibition when using paired-pulse TMS and comparing SCI patients to healthy subjects [50,63]. Moreover, a recent work relying on paired-pulse TMS has measured a specific alteration of MEP amplitude and kinetics that is best explained by diminished capacity to sum up descending excitatory volleys after SCI, rather than by cortical disinhibition [69]. In particular, the authors argue that the most parsimonious explanation for their data was that early corticospinal volleys are insufficient to bring spinal motoneurons to the threshold and that such deficiency is compensated by the later volley. On the other hand, the type of alteration in MEP amplitude and kinetics that was observed in the study was not in line with the expected effects of cortical disinhibition [70,90].

A further example of conflicting outcomes on the matter of cortical disinhibition involves two works based on TMS related to patients with cervical SCI [57,64]. Whereas the first study reports the loss of cortical inhibition as a consequence of SCI [57], the second reveals just the opposite [64]. Remarkably, the second study by Freund and co-workers combines multiple analytical approaches and a larger sample size, resulting in compelling evidence that directly correlates CSP duration to the extent of spinal cord atrophy. It is somehow puzzling to find divergent results in similar studies. However, several factors, including the type of injury, comorbidities, treatments, age and duration of injury, as well as patient demography, motivation and access to support, influence the process of recovery [91], which may contribute to heterogeneity in the course of pathophysiological remodeling of the central nervous system. In this regard, it should also be considered that, besides inter-patient variability, clinical studies on altered excitability and inhibition after SCI are affected by limited accessibility to patients and a lack of options for repeated analyses covering different stages post-injury. Hence, most works focus on few measurements during the chronic phase of SCI (ranging from years to decades after SCI), which is the period providing the highest accessibility from a patient-management perspective. However, it may be revealing to consider what happens to the central nervous system in the very dynamic period shortly after injury, evolving towards the chronic phase [92]. A remarkable study in such a regard analyzes the cortical plasticity of patients from the early weeks post-injury up to a few years after SCI [65]. Interestingly, the study reveals that decreased excitability manifests early (weeks) and lasts for a long time (years) after SCI. However, cortical inhibition is transiently decreased some months after SCI. Hence, an altered balance between excitability and inhibition is the result of alternating events after SCI, and the authors propose that transient disinhibition may be critical for recovery. The unique and novel nature of this work prompts cautious interpretations. Nevertheless, transient loss of inhibition appears once more to be proposed as relevant for motor recovery, as in other earlier studies. Moreover, alternations of phases may explain discrepancies among other works.

Taken together, studies describing altered balances between cortical excitation and inhibition point towards a relevance of this phenomenon in the course of recovery following SCI. However, many confounding factors need to be addressed to improve the comparability of studies made at different clinical centers. Importantly, the severity and position of injury, as well as the time post-injury, need to be considered in the interpretation of TMS outcomes. Moreover, each patient receives a personalized regimen of pharmacological and rehabilitative treatments that has an impact on their cortical plasticity. Therefore, it is necessary for the field to agree on standardized stimulation protocols and methods of analysis that can enable studies with a larger number of patients and adequate controls.

## 6. Possible Causes for the Loss of Inhibition

Pre-clinical research helps to define mechanisms that can converge to an altered ratio of excitation and inhibition following SCI and to refine hypotheses about the reasons for phenotypic heterogeneity, as reported in clinical findings. Focusing on molecular “bottlenecks” of inhibitory neurotransmission, the likely mechanisms of cortical disinhibition after SCI are presented in the following section (Figure 2).

### 6.1. Metabolic Stress

The synthesis of neurotransmitters, such as glutamate and GABA, demand considerable energy consumption [93]. Of these, GABAergic neurotransmission has the highest demand. For this reason, inhibitory neurons such as parvalbumin-positive and axo-axonic interneurons are vulnerable and respond to stress with decreased GABAergic neurotransmission [94]. After SCI, the axotomy of corticospinal neurons may cause stress, spreading from the spinal cord to cortical areas [21,22,23]. Additional sources of stress may be inflammation [95] and impaired cerebral blood flow [96]. Furthermore, stress may coincide with high metabolic demand due to increased cortical excitability [22]. Hence, multiple stressors may affect interneurons, as also suggested by the direct observation of their atrophy after SCI [32]. The alteration in brain network performance as a consequence of dysfunctional interneurons may impact cognition negatively [97,98], such as occasionally documented in SCI patients [99].

### 6.2. Inflammation

Microglia activation plays a pivotal role in SCI. Furthermore, chronic inflammation seems to affect supraspinal regions as well [100]. This is not surprising, given the axotomy of several cortical and subcortical neurons [21,22,23] and the continuity of the spinal cord and brain. Regrettably, although the wave of inflammation after SCI has been well time-resolved near the lesion site [101,102,103], little is known about the spatial distribution of immune cell activation across the CNS following various types of injury and treatment. Thus, it is hard to define which neurons are most exposed to inflammatory processes and which are spared and exclusively undergo plasticity processes independent of inflammation. Among several disinhibitory effects [104], activated microglia affect the activity of proteins that transport chloride across the neuronal cell membrane and cause the dissipation of chloride trans-membrane gradients with consequent reductions in the GABAergic inhibitory drive [105]. Decreased GABAergic drive causes the neurotransmitter GABA to be intrinsically less effective in hyperpolarizing the neuronal membrane to mediate inhibition. The dissipation of chloride gradients upon microglia activation after SCI may therefore contribute to disinhibition as well, documented in other pathologies [105,106].

### 6.3. Remodeling of Perineuronal Nets

The extracellular elements of proteoglycans, known as perineuronal nets, mediate stability in network architecture as well as protection from stress [107]. Dismantling perineuronal nets promotes plasticity and network remodeling [108], which are pertinent for recovery after SCI [109,110]. Furthermore, perineuronal nets are closely associated with specific categories of cortical interneurons [111,112]. Thereby, these extracellular elements can control neuronal synaptic connectivity and intrinsic firing properties [112,113]. Thus, the dismantling of perineuronal nets can affect the activity and connectivity of interneurons, determining altered cortical inhibition after SCI. Strikingly, the advantage of increased plasticity derived from the dismantling of perineuronal nets comes along with the challenge presented by increased metabolic stress. Indeed, perineuronal nets limit excitotoxicity by sheltering synaptic contacts and reducing oxidative stress to which interneurons are most vulnerable [94,114,115]. Since neocortical perineuronal nets undergo remodeling after SCI that is associated with the atrophy of interneurons [32], these extracellular elements may be involved in both beneficial and detrimental aspects of cortical disinhibition.

### 6.4. Altered Astrocyte Metabolism and Physiology

Axotomy and neuronal trauma perturb the physiology of astrocytes [116]. Trauma suffered by cortical principal neurons upon axotomy [21,22,23] may therefore alter signaling from astrocytes to surrounding neurons. Astrocytes can release neuromodulators controlling neuronal activity by gliotransmission [117,118,119]. Moreover, astrocytes are crucial for the metabolism of neurotransmitters [120,121]. Thus, under pathological conditions, altered astrocytic activity may affect neurotransmission, and in particular GABAergic inhibition, via altered metabolic support as well [122]. The involvement of astrocytes in SCI-derived pathology has been widely studied [123], albeit not in cerebral regions.

### 6.5. Rewiring of Cortical Circuits

A physiological process closely tied to disinhibitory mechanisms is the rewiring of cortical circuits after SCI. Like other injuries involving deafferentation of the central nervous system, SCI involves the rewiring of cortical and subcortical areas, as well as the shift of somatotopic representations and changes in competence of motor areas, which relay on significant events of network plasticity [124]. Decreases in GABAergic inhibition appear crucial for network remodeling [125,126] because plasticity and learning are tightly related to changes in cortical GABA [127]. For instance, reduced GABAergic inhibition causes qualitative and quantitative changes in the inducibility of long term potentiation in the neocortex [128,129]. Furthermore, reduced GABAergic tone, i.e., diminished concentrations of extrasynaptic GABA, supports motor plasticity and motor recovery in various types of pathology [126,130]. Thus, several mechanisms that contribute to reduced GABAergic inhibition in the brain after SCI may also partake into the process of cortical rewiring and recovery.

### 6.6. Hyperexcitability of Excitatory Neurons

Some causes of hyperexcitability may be independent from disinhibition. Nevertheless, solving pathological hyperexcitability may necessitate retuning inhibitory components. For instance, axotomy results in the depolarization and hyperexcitability of cortical principal neurons after SCI [22]. The retuning of hyperexcitable neurons may benefit from increased inhibition, for instance, through GABA_B_ receptor signaling, which controls dendritic excitability via intracellular calcium signaling [131]. In contrast, disinhibition has additive exacerbating effects on the intrinsic hyperexcitability of principal neurons. Furthermore, other types of neuromodulation that exert inhibitory control on CNS neurons, e.g., the ones mediated by serotonin, are impaired after SCI [132]. The activation of serotonin receptors has been directly shown to control the excitability of cortical neurons [133]. Additionally, the enrichment of serotonin receptors at the axonal hillock of layer V pyramidal neurons implies a crucial role in the control of cortical functional output [134]. Thus, non-GABAergic neuromodulation can directly control the excitability of principal neurons as well as the excitability of interneurons [135,136]. Since altered serotoninergic neuromodulation can affect directly and indirectly the volume of corticospinal output, serotonin neuromodulation may be a key component in controlling cortical output after SCI. Strikingly, controlling serotoninergic neuromodulation has already proven to support better recovery after SCI [137,138,139].

## 7. Can Therapy Rely on the Loss of Inhibition?

Since disinhibition is integral to plastic rewiring in the central nervous system [124,125,126] and supports motor recovery in pathophysiological conditions [130], it is tempting to hypothesize that therapy after SCI may benefit from actively modulating the balance of excitation and inhibition in cortical and corticospinal networks. Indeed, efforts to improve interventions against SCI have provided encouraging breakthroughs and better understandings of the interconnection between brain plasticity and motor recovery involving the phenomenon of cortical disinhibition.

The repetition of TMS pulses, delivered to the motor cortex at short intervals, modulate corticospinal excitability and improve motor functions, defined as hand movement and dexterity, in incomplete chronic cervical SCI patients [83,140]. The chosen pattern of repetitive stimulation mimicks the pattern of descending late I-waves and reproduces the success of earlier work on incomplete chronic SCI patients [72]. The beneficial effects of such TMS protocols are reported as relying on the modulation of cortical or corticospinal inhibition. Furthermore, TMS protocols of paired associative stimulation are also able to improve corticospinal excitability and provide transient improvements of the motor performance of chronic incomplete cervical SCI patients [141]. In view of these results, the authors argue that the underlying mechanisms rely mainly on processes of long-term potentiation, but a short-term reduction in inhibition facilitates the long-term effects. Regrettably, some studies (e.g., [142]) could not corroborate the effectiveness of repetitive TMS treatments. The authors could not rule out that methodological issues or inter-patient variability contributed to the lack of conclusiveness in their evidence. However, critical differences between studies may also justify different degrees of success. For instance, Kuppuswamy and colleagues treated SCI patients with more heterogeneous types of injury in comparison to the other aforementioned works. Furthermore, Kuppuswamy’s patient medications involved several drugs that affect central nervous system excitability, including an agonist for GABAergic receptors, whereas in more recent work, the administration of drugs known to interfere with cortical excitability was discontinued before the study [141].

Besides TMS-based therapy, periods of loss of cortical inhibition may be useful windows of opportunity for improving recovery based on increased motor exercises. Essentially, lowered interneuron activity in the cortical circuit may, in some SCI patients, become an asset to boost the outcomes of therapy aiming for motor recovery. Following this idea, robotic training successfully achieved increased smoothness and improved aim of movements in chronic cervical SCI tetraplegic patients who had no likelihood of further movement recovery [143]. In such circumstances, motor improvement was monitored with TMS to assess potential changes in the excitability of the central nervous system. Although no direct causative evidence could be provided, the authors explained the effect of the therapy as a modulation of the ratio between excitation and inhibition at the cortical or subcortical level, based on earlier reports on GABA and network reorganization [144].

## 8. Conclusions

Amongst the consequences of SCI on the central nervous system, the loss of inhibition is a common finding, albeit not always observed, and it is likely to fluctuate over time. Changes in cortical excitability involve a plethora of mechanisms, which individual effects may combine in complex and variable outcomes. When reported, the loss of inhibition is mostly proposed to improve motor recovery and can therefore be exploited for that purpose. On the other hand, the loss of inhibition can under some conditions aggravate SCI symptoms in some patients. Thus, the extent of alteration in the balance of excitatory and inhibitory networks should be determined on a patient-to-patient basis and at different times after injury for optimal therapy design. However, this first requires the standardization of assessment methods and the resolution of molecular pathways causing altered cortical dysfunctions in patients after SCI.

## Figures and Tables

**Figure 1 ijms-23-05622-f001:**
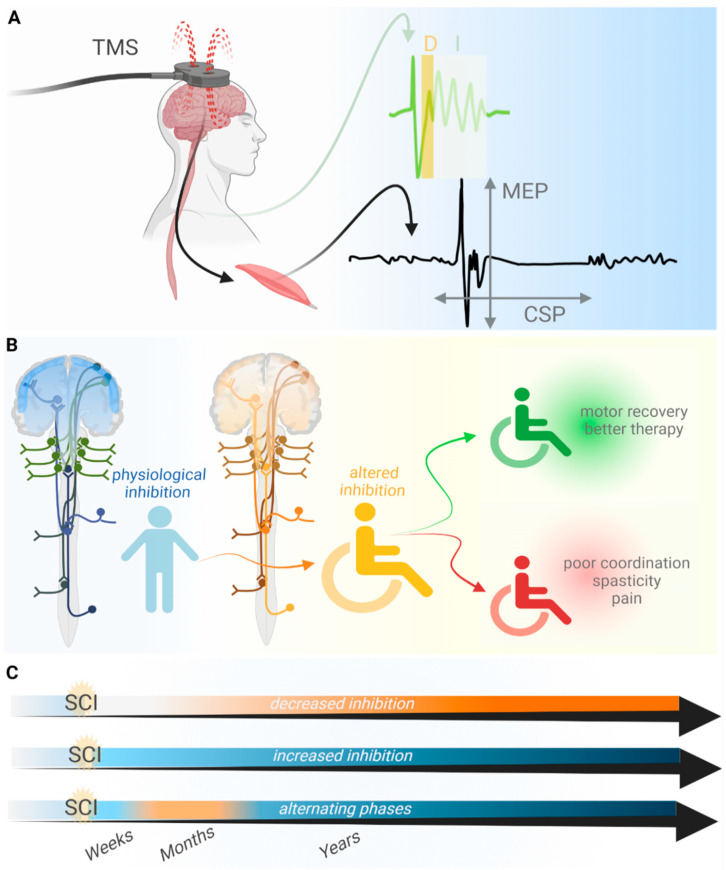
TMS enables measurement of cortical and subcortical excitability and inhibition after SCI. (**A**) TMS-evoked activity reveals the direct and indirect trans-synaptic activation of corticospinal neurons evident as D- and I-waves in descending volleys from the spinal cord. Changes in I-waves reflect altered cortical inhibition after SCI. Similarly, changes in motor evoked potentials (MEP) and cortical silent period (CSP), evoked by TMS pulses and recorded with electromyography, reflect altered integrity of the corticospinal tract after SCI. (**B**) The physiological balance (blue) between excitation and inhibition in cortical and corticospinal networks is perturbed (yellow) by SCI. On one hand, altered inhibition supports rewiring and motor recovery and appears as an exploitable condition in therapeutic treatments. On the other hand, altered inhibition contributes to detrimental aspects, such as exacerbated pain, spasticity and poor motor coordination. (**C**) Altered balance between excitability and inhibition of cortical and corticospinal areas is long-lasting and can endure for decades after SCI. However, alterations vary qualitatively between patients, and therefore, inhibition may be decreased [2,34,57,58,59,60,61,62] or increased [50,63,64] as a consequence of SCI. Moreover, there may be phases of decreased excitability, weeks and years after SCI, interspersed with transiently decreased inhibition months after SCI [65]. Such transient events can contribute to fluctuation in the balance of excitation and inhibition over time during the recovery process. Better resolution of the pathophysiological mechanisms of altered inhibition can allow the determination of relevant factors for the occurrence, duration, heterogeneity and alternation of such phases. This figure was created with biorender.com. Adapted from “Ascending and Descending Spinal Pathways”, by biorender.com. Retrieved from https://app.biorender.com/biorender-templates (accessed on 29 March 2022).

**Figure 2 ijms-23-05622-f002:**
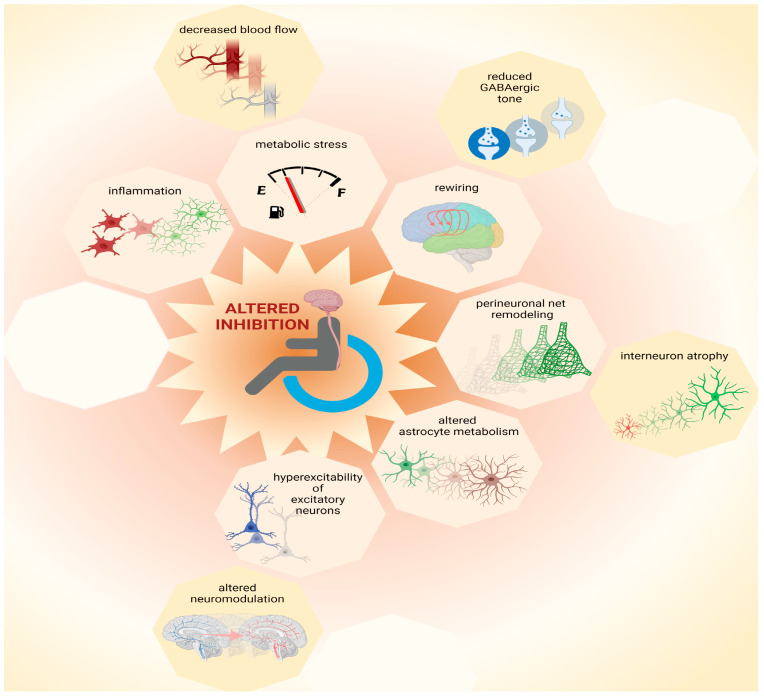
Possible causes for loss of inhibition. Multiple mechanisms contribute to network plasticity and altered inhibition in cortical and subcortical areas after SCI. These include metabolic stress, which can be exacerbated by inflammation and by decreased blood flow, as well as altered astrocytic metabolism. Furthermore, increased excitability of principal (excitatory) neurons may be exacerbated by altered neuromodulation, contributing to increased output volume in corticospinal circuits. Moreover, disinhibition is associated with decreased GABAergic tone that contributes to plasticity during network rewiring. Additionally, remodeling of perineuronal nets, which involves atrophy of interneurons, can contribute to complex patterns of altered inhibition in the central nervous system after SCI. Thus, multiple mechanisms, some of which are yet to be identified (represented by empty octagons) may coexist and combine heterogeneously amongst each other and/or to other pathophysiological components, increasing inter-patient variability. This figure was created with biorender.com.

## Data Availability

Not applicable.

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
