# Peer review of "Spinal Cord Injury and Loss of Cortical Inhibition"

_ijms, 2022, doi:10.3390/ijms23105622_

Round 1

Reviewer 1 Report

The authors summarize the recent advances in the role and exploitation of cortical inhibition during recovery following spinal cord injury (SCI). SCI causes the severing of connections from neurons in the motor cortex to the spinal cord, leading to a loss/deterioration of the voluntary control of movement. Here, they report how transcranial magnetic stimulation (TMS), one of the current therapeutic treatments for SCI, promotes the loss of cortical inhibition with the consequent facilitation of movement. However, the loss of inhibition has also been reported to induce maladaptive changes, like chronic pain. They weight the pros and cons of cortical disinhibition in SCI treatment, identifying the current confounds and possible ways forward.

Major Criticisms

  1. I suggest revising the paragraph “TMS as method to analyse of the loss of inhibition after SCI”, as in the current form, it leaves uncertainty on why TMS is a good method to analyze the loss of cortical inhibition. Having few brief sentences, explaining the concept of cortical disinhibition and the readouts commonly used to detect it, might better introduce the value of TMS as a method. Furthermore, adding a schematic showing the consequences of SCI, the TMS protocol and the changes in cortical excitation/inhibition would be very valuable for the readers.

  1. For a better logic flow, I would suggest to move the Paragraph 6 “Can therapy rely on loss of inhibition?” just before conclusion. This change will help the authors to summarize better what is really the therapeutic potential of cortical inhibition, also in light of the pre-clinical findings, making a better correlation of clinical and pre-clinical studies, and how to synergize efforts.

  1. In section 7 “Possible causes for loss of inhibition” or in the listing of the confounds of understanding excitatory/inhibitory balance, it should be mentioned that there is a major connectivity rewiring in cortical circuits following SCI. This re-wiring could also be the reason of dis-inhibition or maladaptive changes. See for example the review Leemhuis, E.; Giuffrida, V.; De Martino, M.L.; Forte, G.; Pecchinenda, A.; De Gennaro, L.; Giannini, A.M.; Pazzaglia, M. Rethinking the Body in the Brain after Spinal Cord Injury. J. Clin. Med. 2022, 11, 388. https://doi.org/ 10.3390/jcm11020388”.

Minor Criticisms

  1. I recommend editing the paragraph “SCI can occur at any age and damages are irreversible. However, constant improvement of healthcare over the last century significantly ameliorated the quality of life and lifespan of SCI patients…finding a cure, is still elusive [6,7]” to clarify what aspects of SCI recovery have been ameliorated and what are the facets that still require improvement, as in the current formulation it is unclear.

  1. The paragraph “Finding ways to reconnect cortical motor neurons … patient-specific treatment” should be moved after the next paragraph, as for a better flow it would be easier to first explain the loss of connections and then the effort to restore them.

  1. Among the efforts to re-establish corticospinal connections it should also be listed the regeneration approach, see Griffin JM, Bradke F. Therapeutic repair for spinal cord injury: combinatory approaches to address a multifaceted problem. EMBO Mol Med. 2020 Mar 6;12(3):e11505.).

  1. In the paragraph “More recent work agrees with the occurrence of disinhibition after SCI and with its beneficial effects … Once more, disinhibition is portrayed as a rescuing effect contributing to motor recovery.” please check that the reported findings have the relative references (if it is just one paper, please add the ref at the end).

  1. Please edit the sentence “Notwithstanding the wider array of techniques adding … could produce opposite outcomes” to include that the time point of the testing or the TMS protocol used could be potential confounds leading to different outcomes. It would be better to tone down a bit this statement, as it currently leaves the idea that there is no reconciliation possible between all these studies, undermining their validity.

Reviewer 2 Report

In this review, Benedetti et al. summarize recent findings across basic science and clinical findings on the loss of cortical inhibition in spinal cord injury (SCI). This is an interesting topic as the restoration of function for individuals with SCI likely relies on developing a greater understanding of the cortical effects of SCI, and there has recently been significant progress in this domain. The manuscript as a whole is well written, but I have several concerns listed below that need to be addressed. These include inaccurate or misleading citations, the inclusion of figures to summarize the author’s descriptions, and improvements in grammar and syntax.

Major Comments

  • In general, the references used to justify many of the statements in this manuscript seem odd. For example, the first paragraph has the following sentence: “To date, various therapies addressing symptoms of SCI have been developed [3], yet the most ambitious goal, finding a cure, is still elusive [6,7].” The 7th citation makes sense as it is a review focused on SCI recovery. However, the 6th citation is a specific study on the effects of treadmill training and magnetic stimulation on spasticity and gait impairments, not finding a cure. The 3rd citation focuses on one specific therapy (TDCS), not the various therapies being developed. This is just one example, but it is a common issue throughout the manuscript where there are articles cited that likely discuss relevant topics in the cited papers, but are not directly applicable to the statement being made. As this is a review paper, the citations used are vitally important, and therefore, I recommend a thorough review of the references to ensure that they are relevant to each sentence. Further specific information is given in the minor comments.
  • The body of the text is generally well written. However, the abstract has several grammatical errors that need to be remedied. These are detailed in the minor comments.
  • Section 2, Page 2: “Notwithstanding some ambiguities and limitations…” As TMS in SCI is one of the main focuses of this review, these ambiguities and limitations should be discussed in detail.
  • Section 3, Page 3: “More recent work agrees with the occurrence of disinhibition after SCI and with its beneficial effects.” What work? There is only one reference in this entire paragraph, and it does not describe any “beneficial effects”. Furthermore, the sentence “disinhibition is suggested to involve subcortical rather than cortical areas” is not supported by this sole citation. The only sentence in this paper that mentions subcortical areas states “Unmasking of preexisting synaptic connections, owing to disinhibition at cortical or subcortical levels, can be considered the mechanism underlying acute modulation of motor outputs.” This statement does not agree with this sentence and this should be addressed.
  • Section 5, Page 3: “Rather than disinhibition, the underlying mechanisms were proposed to involve modulation by neurotrophins.” The underlying mechanisms of the findings stated in the previous sentence? Neurotrophins are only mentioned once in the discussion of the cited manuscript, and this paper does not discuss paired stimulation at all. This should be revised.
  • The authors discuss in section 5 that the evolution of a SCI from the acute to the chronic phase may contribute to the discrepancies observed in cortical disinhibition. It would be helpful to have a timeline figure showing what is known or hypothesized to occur in cortical disinhibition in the different phases of SCI.
  • The heterogeneity of SCI likely plays a large role in the different results with TMS modulation discussed in section 6. The authors should emphasize the clinical characteristics of the study participants in these studies.
  • A figure summarizing the likely combined effects of the subsections of section 7 would be extremely helpful to explain how cortical inhibition may occur following SCI.

Minor Comments

  • Abstract: “causes permanent deficit” should be “causes permanent deficits”.
  • Abstract: why does the first sentence switch to future tense with “which will correlate”? This should be changed to “which correlates”.
  • Abstract: “Despite of being” should be “Despite being”.
  • Abstract: “physiological activity” can be rephrased to “physiological state”.
  • Abstract: “Conversely” is a little awkward in this context. “However” is more appropriate.
  • Abstract: “Aggravate the long-term outcomes” does not make sense. “worsen the long-term outcomes” is simpler and more straightforward.
  • Abstract: “helps to point out at mechanisms” is awkward. “indicates mechanisms” is more appropriate.
  • Introduction: Why is the reference for the first sentence to a specific study of a drug on motor evoked potentials in rabbits? As this sentence defines what a spinal cord injury is, it would seem more appropriate to cite a general overview of SCI or a clinical description of the effects of SCI. This reference is used again in the following sentence to define further losses of function in SCI such as in bladder, bowel control, and sexual function, but that paper does not even mention those issues.
  • Introduction, Page 1: “However, constant improvement of healthcare over the last century significantly ameliorated the quality of life and lifespan of SCI patients.” This sentence should have a reference.
  • The 3rd citation is to a paper discussing the use of TDCS in SCI, not the effect of rehabilitation on independence and reduction of health complications. Although rehabilitation is discussed in that paper, a more appropriate reference should be made.
  • Introduction: Again, the 4th and 5th citations in the sentence “Regrettably,…” do not actually justify the sentence that recovery remains limited, and are instead focused on robotic training and motor evoked potentials. These papers cite papers that could be used to justify this sentence, and would be much more appropriate.
  • The 8th citation is a response to an editorial response of an article that does not discuss assessing acute SCI severity at all (which is what the sentence discusses). This should be revised.
  • Reference number 23 “Rapid modulation of GABA concentration in human sensorimotor cortex during motor learning” is stated as a paper that studies functional plasticity and reorganization following SCI, but this paper focuses on motor learning in healthy individuals, and does not even mention the word “spinal”.
  • The 25th citation is supposed to support the statement that brain disinhibition has been extensively studied in peripheral deafferentation and stroke, but not SCI. However, this citation investigates the effect of TMS on motor cortex changes in SCI. This should be revised.
  • Section 2, Page 2: “This technique is most versatile as it allows to stimulate…” Use of “this” and “it” should be minimized so that readers do not have to refer to previous sentences. This sentence could be more simply stated as “TMS is used to stimulate the brain safely…”
  • Section 3, Page 2: “Independent evidence converging towards a similar conclusion were obtained by measurement of the cortical silent periods (CSP).” What are the “similar conclusions”? This should be spelled out in this sentence.
  • Section 4, Page 3: “However, there seem to be a flip-side to the story”. This language does not seem appropriate for a scientific publication. “However, there is evidence to suggest that cortical disinhibition is detrimental…” is more appropriate.
  • There is a graphical abstract listed at the end of the manuscript, but I do not see any figures included in the version of the manuscript that I am reviewing. Is this a mistake?

Round 2

Reviewer 1 Report

Nevertheless, the timely implementation of efficient strategies remains of-ten neglected, which affects … decreases the likelihood …”. “which affects” could be edited to “affecting” and “decrease” to “decreasing” for a better flow of the sentence.

Several attempts to address clinical symptoms of SCI have therefore explored the possibility to retune neuronal activity.” It would be clearer to indicate which areas of the central nervous system have been targeted to retune neuronal activity, are authors referring to spinal cord, motor cortex, both, or additional areas? Reference is also needed.

Repetition of TMS pulses, delivered to the motor cortex at short … of incomplete chronic cervical SCI patients [78,138].” The “of incomplete” should be edited to “in incomplete”

Author Response

We thank the reviewer for the further corrections, which we have addressed, and we hope the manuscript now meets your approval.

Nevertheless, the timely implementation of efficient strategies remains of-ten neglected, which affects … decreases the likelihood …”. “which affects” could be edited to “affecting” and “decrease” to “decreasing” for a better flow of the sentence.

Corrected

Several attempts to address clinical symptoms of SCI have therefore explored the possibility to retune neuronal activity.” It would be clearer to indicate which areas of the central nervous system have been targeted to retune neuronal activity, are authors referring to spinal cord, motor cortex, both, or additional areas? Reference is also needed.

We revised the sentence to improve clarity and added some references.

Repetition of TMS pulses, delivered to the motor cortex at short … of incomplete chronic cervical SCI patients [78,138].” The “of incomplete” should be edited to “in incomplete”

Corrected

Reviewer 2 Report

In the revised manuscript, Benedetti et al. have sufficiently responded to my previous comments and addressed the issues appropriately in the text and figures. This has resulted in a manuscript with greatly increased quality. I have included several minor comments and edits that I believe will enhance the readability and understandability of the manuscript.

  • Page 2, Section 2: “TMS is a technique that allows to evoke neuronal activity…” could be simplified to “TMS is a technique that evokes neuronal activity…”
  • Page 2, Section 2: “As consequence, decades of work based on TMS successfully allowed to outline specific traits of altered brain physiology after SCI” could be simplified to “Therefore, decades of work based on TMS successfully outlined specific traits of altered brain physiology after SCI”
  • Figure 1 title: “TMS allows to measure cortical and subcortical excitability and inhibition after SCI” could be edited to improve syntax as “TMS enables measurement of cortical and subcortical excitability and inhibition after SCI”
  • In Figure 1B, is there any difference between the brain/spinal cord diagrams for the “physiological inhibition” and “altered inhibition” categories? It may be helpful to show a difference here to demonstrate alterations in inhibition between the non-SCI and SCI categories. Additionally, it is unclear that altered inhibition is what is simultaneously leading to the “good” and “bad” categories on the right. Indicating that these are coming from the altered inhibition observed due to SCI would greatly aid in the interpretation of this figure. Also, in Figure 1C, if I am understanding the intent correctly, it should be emphasized that the two lines indicate discrepancies in recovery between two cases of SCI and that the orange and blue continue to indicate physiological and altered inhibition consistent with the previous subpanels.
  • Page 3, Section 3: “In this case, the experimental readout was based on measurement of resting motor threshold and motor evoked potentials (MEP) upon TMS stimulation” – The S in TMS stands for Stimulation, and therefore the word stimulation is redundant and can be removed. This also occurs in the first paragraph of Section 5: “Finally, TMS stimulation…”
  • Page 5, Section 5: “Moreover, each patient receives a personalized regimen of pharmacological and rehabilitative treatments that will impact on their cortical plasticity” could be rephrased to “…that will have an impact on their cortical plasticity.” to improve the syntax.
  • Page 5, Section 5: “Therefore, it is necessary for the field to agree on standardized stimulation protocols and methods of analysis that will allow to constitute groups with larger number of patients and adequate controls.” could be simplified to “…that will enable studies with a larger number of patients and adequate controls.”
  • Figure 2 legend typo: “mechanims” -> “mechanisms”
  • Why does Figure 2 have multiple panels such as “decreased blood flow” that are not discussed in the text?
  • Page 7, Section 7: “…it is tempting to hypothesize that therapy after SCI may benefit of actively modulating the balance…”, “may benefit from actively modulating the balance” is more natural syntax.
  • Page 7, Section 7: the use of the word “brave” seems strange. How is the work performed “brave”? Use of superlatives such as this are inappropriate and this should be removed.
  • Page 7, Section 7: “achieved a smoothening” is awkwardly phrased, “achieved increased smoothness” is more straightforward.
  • Page 8, Section 8: “ought to be” -> “should be”

Author Response

We thank the reviewer for the further corrections, which we have addressed, and we hope the manuscript now meets your approval. In the revised manuscript, Benedetti et al. have sufficiently responded to my previous comments and addressed the issues appropriately in the text and figures. This has resulted in a manuscript with greatly increased quality. I have included several minor comments and edits that I believe will enhance the readability and understandability of the manuscript.

Once more, we thank the reviewer for the thorough revision and for the positive criticism. We addressed your comments and increased the quality and clarity of figures.

  • Page 2, Section 2: “TMS is a technique that allows to evoke neuronal activity…” could be simplified to “TMS is a technique that evokes neuronal activity…”

Corrected

  • Page 2, Section 2: “As consequence, decades of work based on TMS successfully allowed to outline specific traits of altered brain physiology after SCI” could be simplified to “Therefore, decades of work based on TMS successfully outlined specific traits of altered brain physiology after SCI”

Corrected

  • Figure 1 title: “TMS allows to measure cortical and subcortical excitability and inhibition after SCI” could be edited to improve syntax as “TMS enables measurement of cortical and subcortical excitability and inhibition after SCI”

Corrected

  • In Figure 1B, is there any difference between the brain/spinal cord diagrams for the “physiological inhibition” and “altered inhibition” categories? It may be helpful to show a difference here to demonstrate alterations in inhibition between the non-SCI and SCI categories. Additionally, it is unclear that altered inhibition is what is simultaneously leading to the “good” and “bad” categories on the right. Indicating that these are coming from the altered inhibition observed due to SCI would greatly aid in the interpretation of this figure. Also, in Figure 1C, if I am understanding the intent correctly, it should be emphasized that the two lines indicate discrepancies in recovery between two cases of SCI and that the orange and blue continue to indicate physiological and altered inhibition consistent with the previous subpanels.

Figure 1 has been edited with attention to details and suggestions, to increase the clarity of its message. The figure legend has been improved accordingly.

  • Page 3, Section 3: “In this case, the experimental readout was based on measurement of resting motor threshold and motor evoked potentials (MEP) upon TMS stimulation” – The S in TMS stands for Stimulation, and therefore the word stimulation is redundant and can be removed. This also occurs in the first paragraph of Section 5: “Finally, TMS stimulation…”

Corrected

  • Page 5, Section 5: “Moreover, each patient receives a personalized regimen of pharmacological and rehabilitative treatments that will impact on their cortical plasticity” could be rephrased to “…that will have an impact on their cortical plasticity.” to improve the syntax.

Corrected

  • Page 5, Section 5: “Therefore, it is necessary for the field to agree on standardized stimulation protocols and methods of analysis that will allow to constitute groups with larger number of patients and adequate controls.” could be simplified to “…that will enable studies with a larger number of patients and adequate controls.”

Corrected

  • Figure 2 legend typo: “mechanims” -> “mechanisms”

Corrected

  • Why does Figure 2 have multiple panels such as “decreased blood flow” that are not discussed in the text?

Items presented in the figure feature in the manuscript, in part 6. However, some items appear as headers and are discussed more extensively and others are only just mentioned along a paragraph. In the figure, we decided to represent all the components comprehensively as colored clusters because we think that it will better serve the readers and engage curiosity. In the improved figure, item names have been made coherent with the text. We hope the current version meets the reviewer’s approval.

  • Page 7, Section 7: “…it is tempting to hypothesize that therapy after SCI may benefit of actively modulating the balance…”, “may benefit from actively modulating the balance” is more natural syntax.

Corrected

  • Page 7, Section 7: the use of the word “brave” seems strange. How is the work performed “brave”? Use of superlatives such as this are inappropriate and this should be removed.

Removed

  • Page 7, Section 7: “achieved a smoothening” is awkwardly phrased, “achieved increased smoothness” is more straightforward.

Corrected

  • Page 8, Section 8: “ought to be” -> “should be”

Corrected